# Integrated Analysis of mRNAs and Long Non-Coding RNAs Expression of Oviduct That Provides Novel Insights into the Prolificacy Mechanism of Goat (*Capra hircus*)

**DOI:** 10.3390/genes13061031

**Published:** 2022-06-08

**Authors:** Zhipeng Sun, Zijun Zhang, Yufang Liu, Chunhuan Ren, Xiaoyun He, Yanting Jiang, Yina Ouyang, Qionghua Hong, Mingxing Chu

**Affiliations:** 1Key Laboratory of Animal Genetics, Breeding and Reproduction of Ministry of Agriculture and Rural Affairs, Institute of Animal Science, Chinese Academy of Agricultural Sciences, Beijing 100193, China; fsunzhipeng@163.com (Z.S.); aigaiy@126.com (Y.L.); hexiaoyun@caas.cn (X.H.); 2College of Animal Science and Technology, Anhui Agricultural University, Hefei 230036, China; zhangzijun@ahau.edu.cn (Z.Z.); renchunhuan@126.com (C.R.); 3Yunnan Animal Science and Veterinary Institute, Kunming 650224, China; jiangyanting-2007@163.com (Y.J.); yinaouyang@163.com (Y.O.)

**Keywords:** goat, oviduct, prolificacy, lncRNA–mRNA, pathways, follicular phase

## Abstract

Artificial directional selection has replaced natural selection and resulted in trait differences across breeds in domestic animal breeding. However, the molecular mechanism by which the oviduct regulates litter size remains largely elusive in goats during the follicular phase. Accumulating data have linked lncRNAs to reproductive activities; however, little is known about the modulation mechanism in the oviduct. Herein, RNA-seq was used to measure mRNA and lncRNA expression levels in low- and high-fecundity goats. We observed distinctive differences in mRNA and lncRNA in terms of different kidding numbers and detected the differential expression of 1640 mRNA transcripts and 271 lncRNA transcripts. Enrichment analysis of differentially expressed mRNAs (DEGs) suggested that multiple pathways, such as the AMPK, PI3K–Akt, calcium signaling pathway, oocyte meiosis, ABC transporter, and ECM–receptor interaction pathways, directly or indirectly affected goat reproduction. Additionally, coexpression of differentially expressed lncRNAs (DEL)-genes analysis showed that XLOC_021615, XLOC_119780, and XLOC_076450 were *trans*-acting as the DEGs *ATAD2*, *DEPDC5*, and *TRPM6*, respectively, and could regulate embryo development. Moreover, XLOC_020079, XLOC_107361, XLOC_169844, XLOC_252348 were the *trans*-regulated elements of the DEGs *ARHGEF2* and *RAPGEF6*, and the target DEGs *CPEB3* of XLOC_089239, XLOC_090063, XLOC_107409, XLOC_153574, XLOC_211271, XLOC_251687 were associated with prolificacy. Collectively, our study has offered a thorough dissection of the oviduct lncRNA and mRNA landscapes in goats. These results could serve as potential targets of the oviduct affecting fertility in goats.

## 1. Introduction

Goat farming is practiced worldwide, with goat products having a favorable image [1]. Goat production is one of the key elements contributing to the economy of farmers living in arid and semiarid regions [2,3]. The number of goats has increased globally, even in countries with high and intermediate incomes [4]. Worldwide, 96% of the milk- and meat-producing goat populations are found in developing countries, and 4% are found in developed countries [5,6]. Mammalian prolificacy is controlled by a complete reproductive system; as one of the most important organs, the oviduct is directly involved in the reproductive process. Of note, it contributes to selecting sperm, determining the numbers of sperm reaching the site of fertilization, storing and transporting gametes, and the initial stages of embryonic development, thereby controlling polyembryony [7,8,9,10]. During the differentiation–estrus cycle, the morphology and distribution of oviduct cell types are distinct, and understanding the molecular mechanisms of the oviduct in the follicular phase may help us reveal the differences in litter size [11,12]. For example, in the follicular phase, multiciliated cells (MCCs) are highly enriched in the ampulla (the site of fertilization) of the oviduct, which is near the ovary and responsible for capturing oocytes [13]. This provides a new ideal for understanding the prolificacy of nanny goats from the perspective of the oviduct.

The molecular breeding method can significantly shorten the breeding cycle compared with the traditional method, and it allows for more purposeful breeding of new varieties with targeted traits, such as high fertility. For the past few years, much research on the mechanisms of litter size in goats has focused on genes, non-coding RNAs, signaling pathways, and transcription factors [14]. Recent studies have shown that the expression of oviduct genes is also associated with fertility. In both humans and mammals, the oviductal *glycoprotein 1 gene* (*OVGP1*) has positive effects on sperm, ovum infiltration, fertilization, and early embryonic development [15,16]. Acuña OS et al. described *GPX2* in the oviduct and suggested that this protein may protect gametes (oocytes and sperm) from oxidative damage as they pass through the oviduct [17]. *MUC1*, expressed in the human fallopian tube, is a key factor in ectopic pregnancy [18]. Nevertheless, these genes in the oviduct have different effects on prolificacy in the different estrus cycles. Therefore, it is essential to identify other major genes and regulators in the follicular phase of the oviduct that affect goat prolificacy.

A comprehensive analysis of previous studies highlights that non-coding RNAs (ncRNAs) act as eukaryotic gene regulators. LncRNAs are one of the highly expressed ncRNAs that are usually identified as having a length of more than 200 nt and impact the regulation of neighboring gene expression in numerous important biological processes [19,20]. LncRNAs have been reported to engage the epigenetic or transcriptional regulation of neighboring genes as cis or trans-elements [21]. This type of regulatory capacity can be demonstrated in several reproductive processes, such as LOC102191297 and LOC102171967 targeting *SRD5A2* in the steroid hormone biosynthesis pathway to regulate goat reproduction [22], and Lnc5926 targeting *ZSCAN4* and *EIF1AX*, which are essential for early embryonic development [23]. Importantly, numerous works have proved that lncRNA expression can lead to early germ cell formation and embryo implantation, and the process of mammalian animal reproduction [24,25]. In goats, lncRNAs have been identified in the ovaries, uterus, and germ cells [26,27,28] and might regulate the oogenesis and maturation, embryo implantation, and oocyte follicle development. Wang et al. analyzed lncRNAs and mRNAs in goat perirenal adipose tissues and found that 4519 lncRNAs and 5212 mRNAs were potentially in trans-regulatory relationships and that the target genes of lncRNAs were associated with brown to white adipose tissue transformation [29]. In addition, a novel study that performed RNA-seq identified 20 DELs of high-fecundity goat ovaries during estrus, and the lncRNAs that regulate goat fecundity were predicted [30]. However, little is known about the exact role of lncRNAs in the oviduct, especially in the molecular regulation of litter sizes.

Increasing meat production using scientific, accurate, and precise selective programs is one of the most important goals for the genetic improvement of goats [31,32]. The Yunshang black goat with high litter size was the first mutton black goat bred in China. However, the molecular mechanism of prolificacy remains unclear. Elucidating the transcriptome expression profile of the oviduct in goats is important for understanding the molecular mechanisms by which the oviduct regulates litter size. It can not only screen out candidate coding genes related to doe fertility but also identify key reproductive regulatory factors of the oviduct. In our study, RNA-seq was used to identify the differential expression of mRNAs and lncRNAs expression in the oviducts of high- and low-fertility goats. Oviduct tissues were collected from five low-fecundity and five high-fecundity goats in the follicular phases. To understand the relationship between mRNA, lncRNA, and reproduction, DEGs and DELs were screened for bioinformatics analysis. We also established a regulatory network that targeted relationships between these hub DELs and target genes (DEGs). This study systematically presented characteristics of goat oviduct lncRNA and mRNA profiles that could be used as specific biomarkers of goat fertility and oviduct function. Therefore, our results might provide basic data for Yunshang black goat prolific trait-related lncRNAs and candidate genes and contribute to molecular breeding.

## 2. Materials and Methods

### 2.1. Ethics

The protocols for the animal study were reviewed and approved by the Institute of Animal Sciences, Chinese Academy of Agricultural Sciences, Beijing, China. Ethical approval was given by the Animal Ethics Committee of the IAS-CAAS (No. IAS2021-23). Written informed consent was obtained from the owners for the participation of their animals in this study. All animal handling was carried out under the license approved by Yunnan Animal Science and Veterinary Institute.

### 2.2. Animals and Sample Collection

Adult doe, with the same weights and hearths, were raised with free access to feed and water under the same temperature. They were obtained from the Yixingheng Animal Husbandry Technology Co., Ltd. Tuanjie Township Base in Kunming City (Yunnan, China). In this research, 10 selected goats (3 years) were divided into two groups (n = 5 for each group): the low-fertility group (average litter size 1.8 ± 0.27, FL) and the high-fertility group (average litter size 3.4 ± 0.42, FH) according to the litter size records. They were treated with simultaneous estrus based on progesterone vaginal suppository (CIDR) during the nulliparous period. Then, the CIDR was removed after 16 days and all goats were slaughtered (euthanized: intravenous pentobarbital 100 mg/kg) at 48 h (follicular phase), and whole oviduct samples were collected from ten goats individuals, respectively. Samples were quickly stored in RNAlater reagent and deep-frozen in liquid nitrogen and stored at −80 °C for RNA extraction.

### 2.3. RNA Isolation, Library Preparation, and Sequencing

According to the manufacturer’s protocol, TRIzol reagent (Invitrogen, Carlsbad, CA, USA) was used to extract the mRNA from frozen oviduct samples of 10 goats (mixed powders of the entire oviduct) for RNA-seq. The RNA concentration (total RNA ≥ 3000 ng/µL) and purity (OD 260/280: 1.8~2.2) were measured using a NanoDrop^TM^ 2000 (Thermo Scientific^TM^, Wilmington, DE, USA) instrument, and RNA integrity was evaluated using an Agilent 2100 System (Agilent Technologies, Santa Clara, CA, USA). RNase-Free 1.5% agarose gel electrophoresis was used to detect whether the RNA sample was disseminated and whether genomic DNA was contaminated.

We collected 10 RNA samples from oviduct tissues for RNA-seq. Ten mRNA and lncRNA libraries were prepared using Illumina’s TruSeq library preparation kits following the protocol recommended by the manufacturer. Briefly, mRNAs with poly-A tails were enriched by Oligo (dT) magnetic beads. Then, rRNAs (ribosomal RNA) were depleted by the Ribo-Zero^TM^ rRNA Removal Kit (Epicenter, San Antonio, TX, USA) by digesting the total RNA, and rRNA-free residues were cleaned by ethanol precipitation. Next, the NEBNext^®^ UltraTM Directional RNA Library Prep Kit and Illumina (NEB, Ipswich, MA, USA) were used to generate mRNA sequencing libraries, and lncRNA libraries were generated by strand-specific according to the manufacturer’s recommendations for each sample. Finally, the cDNA of approximately 250–300 bp was screened with the AMPure XP beads system, the PCR products were purified again with AMPure XP beads to obtain the libraries, Qubit 2.0 was used for accurate quantification, and the Agilent Bioanalyzer 2100 system (Agilent Technologies, Santa Clara, CA, USA) was used to evaluate library quality. Subsequently, the libraries were sequenced on the Illumina NovaSeq platform (150 bp paired-end reads, Illumina, Hayward, CA, USA) at Wuhan Frasergen Bioinformatics Co., Ltd. (Wuhan, China).

### 2.4. Quality Control and Reference Genome Mapping

Fast-QC (http://www.bioinformatics.babraham.ac.uk/projects/fastqc/, accessed on 5 January 2022) was used to check the quality of the raw RNA-seq (both the mRNA-seq and lncRNA-seq) data. The raw reads in fastq format were first filtered by SOAPnuke (v2.1.0) using the criteria for removing low-quality reads with splice sequences, poly-N bases accounting for more than 1% of reads, and filtering out the reads that formed paired-end pairs with low quality. The Phred quality score represented the rate of different base sequencing errors, such as Q20 and Q30, indicating that the base sequencing error rates were 1% and 0.1%, respectively. Then, clean Illumina sequencing reads were retrieved, among which the number of bases with a quality value Q ≤ 20 was >50%. The Q20, Q30, and GC contents of the clean data were calculated. Immediately, HISAT2 (v2.1.0) [33] was used to map the mRNA-seq clean reads to the reference genome (the reference genome and gene model annotation files were downloaded directly from the genome website). Subsequently, we used String Tie (v1.3.5) [34] software to assemble transcripts of mRNA. Both known and novel mRNA from the String Tie (v1.3.5) alignment results were constructed and identified by the Cufflinks (v2.1.1) Reference Annotation Based Transcript (RABT) assembly method. The fragments per kilobase per million reads (FPKM) [35] were calculated based on the length and read counts of each mapped gene.

### 2.5. lncRNA Identification and Differentially Expression Analysis

The lncRNA-Seq data were based on transcriptome splicing results. Putative lncRNAs, including lincRNA, antisense-lncRNA, and intronic-lncRNA, were identified with the following steps. First, only transcripts with length ≥ 200 bp, reads > 3, and exon number > 1 were retained. Second, Cuffcompare was used to discard transcripts of mRNAs and other non-coding RNAs (rRNA, tRNA, snoRNA, etc.) in the known database, and transcripts annotated as “i,” “j,” “o,” “u,” and “x,” representing lncRNAs of potentially novel introns, potentially novel isoforms with more than one splice junction of a reference transcript, generic exonic overlaps with a reference transcript, potentially intergenic sequences, and potentially antisense transcripts, respectively, were kept. Finally, the transcripts without coding potential were predicted by three coding potential prediction software programs, CNCI (Coding-Non-Coding Index) [36], CPC2 (http://cpc2.cbi.pku.edu.cn/, accessed on 6 January 2022) [37], and PLEK [38], and the intersections of these prediction results were defined as novel lncRNAs. CNCI was used to distinguish between the coding and non-coding potential of sequences, by analyzing the spliced transcripts based on the profiling of adjacent nucleotide triplets, with the parameter of score < 0. CPC2 was used to determine the protein-coding potential of each transcript, according to the transcript sequences in Fasta format, and was divided into coding and non-coding transcripts. PLEK was used to distinguish coding probability by the kmer composition of the sequences without comparison, and a score <0 indicates that the lncRNA transcript was defined as a non-coding RNA. As a result, the statistics of lincRNA, antisense-lncRNA, and intronic-lncRNA statistics were obtained.

The mRNA and lncRNA expression levels in both groups (with five biological replicates) were estimated by FPKM values, which eliminated the effects of sequencing depth, gene length, and sample difference on gene expression levels. The expression levels of genes and lncRNAs at the follicular phase of LL were compared with the LH group, and the DESeq R package (1.8.3) was used to calculate log_2_(fold change) and *p*-value based on the normalized counts [39]. Furthermore, lncRNAs and mRNAs with threshold values of *q*-value ≤ 0.05 (the *q* values were adjusted by the *p*-values) and |log_2_ FC (fold change)| ≥ 1 were considered as differentially expressed (DE) between the low- and high-fertility groups.

### 2.6. Potential Target Gene Prediction and Networks Construction

We predicted the relationships between lncRNAs and potential protein-coding genes according to their distances and coexpression. Protein-coding genes with 100 kb upstream and downstream of the candidate lncRNAs were identified as potential *cis*-elements and Pearson correlation coefficients with the lncRNAs of greater than 0.95 were identified as the potential trans-acting target genes of the DELs [20]. Then, the target genes were integrated with DEGs in the profile. A protein–protein interaction (PPI) network was constructed by the STRING database (https://string-db.org/, accessed on 10 April 2022) and used to infer the relationships of DEGs with different groups (FL vs. FH). Furthermore, we built a lncRNA–mRNA network based on the targeting relationship, and the network of lncRNAs along with their target genes was visualized using Cytoscape software (V3.9.0). Cytoscape was run with “layout = ‘attribute circle layout’”, with the other parameters set as the default.

### 2.7. Function Enrichment Analyses

To uncover the key roles of DEGs and DELs, GO (Gene Ontology) and KEGG (Kyoto Encyclopedia of Genes and Genomes) pathway enrichment analyses were performed using the online software Metascape (https://metascape.org/gp/index.html#/main/step1, accessed on 15 April 2022) with default parameters to explore the potential functions of all DGE and DEL target genes. GO terms were classified into cellular components (CC), molecular functions (MF), and biological processes (BP). KEGG is a central public database for understanding genomic information with functional information. Next, only the enriched GO terms were considered significant with an adjusted *q*-value (*t*-test) < 0.05, and KEGG pathways were considered significant with a *p*-value (*t*-test) < 0.05.

### 2.8. Quantitative Real-Time PCR Validation of lncRNA and mRNA

The previously snap-frozen oviduct tissues were minced using a homogenizer (Genenode, Beijing, China), and RT-qPCR has performed to validate the accuracy of sequencing data. Briefly, total RNA was extracted from these milled samples using TRIzol (Invitrogen, Carlsbad, CA, USA), following the manufacturer’s protocol. RNA integrity and concentration were assessed by denaturing agarose gel electrophoresis and NanoDrop 2000 spectrophotometry (OD260/280 ratios between 1.95 and 2.0), respectively. For real-time quantitative PCR (RT-qPCR) analysis, cDNA (complementary DNA) was performed from reverse transcription of 1000 ng RNA using the PrimeScript^TM^ RT reagent kit (TaKaRa, Beijing, China). The primers of target genes and lncRNAs were designed using the software Primer Primer 6 and synthesized by Sangon Biotech Co., Ltd. (Beijing, China), and PRL19 was used as the reference gene. RT-qPCR was performed using the TB Green^®^ Premix Ex Taq ^TM^ II (TaKaRa, Beijing, China) according to the manufacturer’s instructions and all genes and lncRNAs were performed in triplicate, and then the qPCR was analyzed on the QuantStudio^®^ 3 (ABI, Foster City, CA, USA). A total of 5 mRNAs and 4 lncRNAs were selected for validation via RT-qPCR, and all oligo sequences of primer are listed in Appendix A. In conclusion, the data obtained from the RT-qPCR reaction were calculated with the 2^−ΔΔCt^ method [40]. The results were analyzed by SPSS 20.0 software (SPSS Inc., Chicago, IL, USA) and were displayed as “means ± SEM” of five replicates. T-tests was performed to compare means, and *p*-value < 0.05 was considered statistically significant (* *p* < 0.05; ** *p* < 0.01).

## 3. Results

### 3.1. Overview of Transcriptome Sequencing Data

To identify the putative transcripts in the goat oviduct, 10 oviduct samples were obtained from Yunshang black goats at follicular phases, and RNA libraries were constructed by the Illumina NovaSeq platform for RNA-seq. The cDNA fragment lengths were immediately selected to be 150 bp. More than 50 million clean reads were obtained in each group after filtering low-quality reads from raw reads. The average percentages of Q 20 and Q 30 were 98.26% and 94.65%, respectively, and the GC content was more than 48.55% for high-quality reads. The information is listed in Table 1. To verify the reliability of the sequencing results, HISAT2 [33] was used to compare and analyze the clean reads with respect to the reference genome. There were 96.40~97.35% of the total mapped reads and 88.36~90.93% of unique mapped reads that were aligned to the reference genome, and the multiple mapped reads were less than 8.56%, which showed that the sequencing data were adequate for further analysis (Table 1).

### 3.2. Identification of mRNA and lncRNA in YunShang Black Goat Oviduct Tissue

According to the sequence data, a scatter plot was applied to display the difference in expressed mRNAs between the FL and FH groups (Figure 1A). A total of 68,149 mRNA transcripts, including 43,779 protein-coding mRNAs and 24,370 novel mRNAs (Appendix A), were identified in 10 oviduct tissues. Regarding the mRNA expression levels in the 10 oviduct tissues, only 1.14% of genes were highly expressed, with the FPKM (the number of uniquely mapped fragments per kilobase of exon per million fragments mapped) greater than 60, and the expression levels of most genes with FPKM were less than 1 (Appendix A). Furthermore, we analyzed the expression levels with Log_10_ (FPKM+1) between lncRNAs and genes, and the results showed that the mRNA transcript expression levels were higher than the lncRNA transcripts expression levels in the oviduct, as shown in Figure 1B. Combining the Cuffcompare classes, 1788 known lncRNAs were identified, and according to the CNCI, CPC2, and PLEK coding potential prediction software, 24,170 common novel lncRNAs were selected (Figure 1C, Appendix A). The majority of novel lncRNAs included 13.64% antisense-lncRNAs (3297), 36.48% lincRNAs (8817), and 49.88% intronic-lncRNAs (12,056) (Figure 1G, Appendix A). These lncRNAs were not uniformly distributed on the 29 autosomes and X-chromosomes; the chromosome distribution of lncRNAs showed that chromosome 3 contained 5.28%, followed by chromosome 5 (4.88%) and chromosome 18 (4.75%) (Figure 1E). In addition, whole lncRNA transcripts and mRNA transcripts were indicated to have originated from protein-coding exons of the gene. In this study, most of the lncRNAs contained only 2–6 exons, while mRNAs had a wide range of exons (from 1 to 40) and were significantly more abundant than lncRNA transcripts (Figure 1F, Appendix A). As Figure 1D shows, the length of the lncRNA was mainly distributed in the 200–600 bp range, while most of the mRNAs ranged from 200 bp to 4000 bp.

### 3.3. Analysis of DEG and DEL Expression Profiles in the Goat Oviduct

Under the criteria of |log_2_ FC (fold change)| > 1 and *q*-value < 0.05 normalized expression, 1640 DEG transcripts (622 genes) were identified (818 DEGs were upregulated and 822 DEGs were downregulated) (Figure 2A, Appendix A), and 271 DEL transcripts were screened (135 DELs were upregulated and 136 DELs were downregulated) (Figure 2B, Appendix A) in the comparison of the low- and high-fertility groups. Then, we performed a clustered heatmap, a visualization method, and analyzed differentially expressed lncRNAs and differentially expressed mRNAs to better explore whether the grouping was reasonable, and heatmaps were constructed, as shown in Figure 2C (Appendix A). We carried out an analysis of the differentially expressed mRNAs; notably, the *TET3*, *PRLR*, *FGFRL1*, *ATP2A3*, *CAMK2G*, and *PXN* were related to reproduction.

### 3.4. PPI Analysis of the DEGs Related to Prolificacy Trait

To preliminarily evaluate the functions of DEGs, we focused on the posttranslational protein levels of these differentially expressed mRNAs. Depending on the minimum required interaction score (0.9), protein–protein interaction networks (PPIs) were established using the Search Tool for the Retrieval of Interacting Genes (https://string-db.org/STRING, accessed on 10 April 2022). There were 131 nodes and 107 edges in the network, and *BRCA1*, *PXN*, *GLI1*, *PRKAG2*, *JKA1*, *CAMK2G*, *IRF4*, *IRF5*, *RSAD2*, *KLHL12*, *KLHL7*, *MCM9*, *GINS3*, and *TIMELESS* were the key degrees of the networks with other proteins (Figure 3).

### 3.5. Functional Analysis of the DEGs

To predict the functions of DEGs and the target genes of DEL, GO terms and KEGG pathway analyses were performed. The DEGs were enriched in GO terms by analyzing the function through biological processes (BP), molecular function (MF), and cellular components (CC). As a result, a total of 269 GO terms (*q*-value < 0.05, Appendix A) were significantly enriched, of which 156, 41, and 72 terms were enriched in the terms BP, MF, and CC, respectively. The top terms in the biological process category included actin filament-based process, regulation of plasma membrane-bounded cell projection organization, regulation of cell projection organization, actin cytoskeleton organization, and small GTPase-mediated signal transduction. In the cell components, the most highly enriched terms were cell leading edge, cell projection membrane, collagen-containing extracellular matrix, extracellular matrix, and external encapsulating structure. In the molecular functions, GTPase regulator activity, nucleoside-triphosphatase regulator activity, GTPase activator activity, phosphotransferase activity, alcohol group as acceptor, and kinase activity were significantly annotated. The top 20 GO terms of each type of function are shown in Figure 4A (Appendix A).

Next, KEGG enrichment analysis was performed to identify the possible pathways of the DEGs. The results showed that 44 basic pathways (*p*-value < 0.05) were found to be enriched, including AMPK, PI3K–Akt, Jak–STAT, cGMP–PKG, Hippo, HIF-1, PPAR signaling pathway, ECM–receptor interaction, ABC transporters, focal adhesion, regulation of actin cytoskeleton, and Th17-cell differentiation (Figure 4C). Furthermore, KEGG pathway analysis revealed that circadian rhythm-mammal and oocyte meiosis related to reproduction were enriched. These pathways may be involved in the regulation of oviductal function in the goat at the follicular phase. In addition, several DEGs were involved in multiple pathways simultaneously, such as *FN1* and *COL4A5*, which functioned in four pathways, *ADRA1A* and *MTOR* participated in three pathways, and *BRCA1* and *VDAC3* were enriched in two pathways

### 3.6. Analysis of the Targeting Relationship between lncRNAs and mRNAs

The differentially expressed mRNAs and lncRNAs with Pearson correlation coefficients above 0.95 were selected to construct the network of lncRNAs–mRNAs using Cytoscape v.3.9.0. Our results showed that 370 nodes and 2858 lncRNA–mRNA pairs were included in this network. Interestingly, the known lncRNA–KLC1 was the cis-regulated element of *BAG5*, and the lncRNA–LOC106502761 targeted 14 protein-coding genes in a *trans*-acting mechanism, while 138 novel lncRNAs corresponded to 230 target protein-coding genes (Appendix A). It was shown that lncRNAs and mRNAs were mutually regulated in the oviduct. In addition, the lncRNA-associated mRNAs were categorized in the GO terms and KEGG pathways to understand the functions of these lncRNAs and the target genes. In the comparison of the FL vs. FH groups, the target genes were enriched in GO terms, including cellular component morphogenesis, cell junction organization, microtubule-based transport, axonal transport, GTPase regulator activity, actin-binding, and other subclasses (*q*-value < 0.05) (Figure 4B, Appendix A), and KEGG pathways, axon guidance, PI3K–Akt signaling pathway, PPAR signaling pathway, ABC transporters, adherens junction, regulation of actin cytoskeleton, MAPK signaling pathway, cytokine–cytokine receptor interaction, tight junction, homologous recombination, and neuroactive ligand–receptor interaction were significantly annotated and are visualized in Figure 4D (*p*-value < 0.05) (Appendix A). These results showed that lncRNAs can interfere with target genes to regulate the oviduct functions of reproduction. Since pathway analysis is significant for understanding the functions of these target genes, we selected some KEGG pathways related to the function of the oviduct, such as the AMPK, PI3K–Akt, and ECM–receptor interaction pathways, and associated mRNAs and lncRNAs were used to create the partial coexpression network by applying Cytoscape (Version 3.9.0). There were 91 novel lncRNAs corresponding to 31 target genes through *cis*- or *trans*-acting pathways (Figure 5). The functions of some genes were regulated by lncRNAs in different pathways. For example, when we focused on the ABC transporter pathway, eight target genes (*PDPK1*, *PLXNB3*, *PTCH1*, *TRPC3*, *SRGAP3*, *ABLIM3*, *ENAH*, and *PARD3*) were regulated by lncRNAs to function in this pathway, while these DEGs were involved in other pathways. Nevertheless, three target genes (*ABCA3*, *ABCA6*, and *ABCC5*) were involved in the ABC transporter pathway, regulated by lncRNAs, and these DEGs play roles in the axon guidance pathway. Among these target genes, only *PDPK1* is separately regulated by XLOC_212128 through a *cis*-acting mechanism, and other genes are jointly regulated by multiple lncRNAs.

### 3.7. Verification of RNA Expression Profiles with Quantitative Real-Time PCR

To identify mRNA and lncRNA changes in abundance and validate the accuracy of the RNA-seq results, depending on relatively high abundance with *q*-value ≤ 0.05 and |log_2_(fold change)| ≥ 1, 5 DEGs (*MAPK6*, *FN1*, *FAM107A*, *JAK1*, *CXCL14*) and four novel DELs (XLOC_163069, XLOC_250759, XLOC_236742, XLOC_031924) were randomly selected to validate their expression using RT-qPCR validation. Compared with the RNA-seq data, the results showed consistent expression patterns, suggesting that the RNA-seq data are highly accurate (Figure 6).

## 4. Discussion

Clarifying the regulatory mechanism of the kidding number can provide a dynamic theoretical basis for goat breeding. Although the ovary determines the number of ovulations, the oviduct, as a conduit between the ovary and uterus, plays important roles in gamete storage, maturation, fertilization, and early embryonic development [41,42,43]. It largely determines the success of reproduction in mammals. Previous studies on oviduct function have been carried out in humans [41], bovines [44], and porcines [17], but they are limited to the study of mRNA. Nonetheless, the roles of lncRNAs in oviduct reproduction remain largely unknown. Therefore, the data mining of critical genes and the regulatory mechanisms of non-coding RNAs in goat prolificacy traits is still a large research topic. Herein, RNA-seq was used to analyze the transcriptome expression profiles of oviduct mRNA and lncRNA in Yunshang black goats with low and high fertility during the follicular phase.

In this study, the functions of lncRNA and the target genes of lncRNAs were revealed from the perspective of the oviduct in the follicular phase. A total of 68,149 mRNAs and 25,958 lncRNAs were identified in the goat oviduct, of which 24,170 novel lncRNAs and 1788 known lncRNA transcripts were obtained. The results of sequence analyses showed that the GC content was 48.55~51.45%, whereas the Q30 content was 93.50~95.25% by the base quality and composition analysis, indicating that high-quality libraries and sequencing were applied. Many studies have shown that lncRNAs have unique characteristics compared with mRNAs; for example, the transcripts of lncRNAs tend to be shorter in length, have fewer exons, and have lower expression levels [45,46]. These characteristics were consistent with this research. In addition, compared with the 4926 and 11,723 lncRNAs found in the ovary [26] and uterus [27] of goat, respectively, there are more lncRNAs in the oviduct, but their exons are similar, which reflects the specific expression patterns and characteristics of the oviduct. At the same time, most lncRNAs were widely located on chromosomes 3 (NC_030810.1), 5 (NC_030812.1), and 18 (NC_030823.1), which is inconsistent with the distribution in other reproductive organs of goats. All the results explained that lncRNAs in particular were reliable in the oviducts of goats.

Herein, we identified a wealth of GO terms and KEGG pathways. Subsequently, we analyzed the important pathways and pertinent DEGs, which are involved in reproductive regulation, including AMPK, PI3K–Akt, cGMP–PKG, HIF-1, Jak–STAT, and calcium signaling pathway, ECM–receptor interaction, ABC transporters, protein digestion and absorption, focal adhesion, tight junction, circadian rhythm, oocyte meiosis, and ubiquitin-mediated proteolysis. Reproduction is a function that requires energy and can only be carried out when mammals have enough available energy. The AMPK signaling pathway serves as a cellular energy receptor that responds to low levels of ATP, and AMPK activation affects gene expression in biological processes such as embryonic development, spermatocyte DNA damage repair, and gonadal steroid hormone production, suggesting that AMPK activation affects the transcriptional regulation of reproductive processes [47]. The follicular phase is a key period of follicular development, and the normal development of follicles and ovulation is an important reason for high fecundity. In the reproductive system, the PI3K–Akt signaling pathway plays important roles in regulating cell metabolism, as well as in the survival and activation of primordial follicles [48,49]. The oviduct plays active roles in the processes of oocyte maturation, migration, and fertilization, which are important in reproduction and can directly determine the kidding number. Some studies have pointed out that the interaction between oocytes and oviducts in mammals has a positive impact on oocyte maturation and can restore the oocyte meiosis process [50]. Notably, the focal adhesion and tight junction pathways were significantly enriched in the oviduct in the follicular phase, which are important functions that determine fertilization [51]. Our study showed that these DEGs, including ABCC9 and APC, were upregulated in the FL vs. FH groups and were involved in ABC transporters and regulation of the actin cytoskeleton, respectively. It has been reported that *ABCC9*, a molecule that matches the K (+)-selective pores, regulates the ABC transporter protein of K (+) channels by assembling ATP-sensitive K (+) channels in smooth muscle and transverse muscle [52]. Fertilization occurs primarily in the ampulla of the oviduct, which has an extensively folded epithelial layer and is collated with a thin layer of smooth muscle [43]. Hence, we hypothesize that *ABCC9* contributes to the contraction of oviductal epithelial tissue muscles, thereby aiding gamete transport and contributing to complete fertilization. Meanwhile, *APC* can promote cell proliferation, differentiation, migration, and apoptotic [53]. In the oviduct, the calcium signaling pathway plays an important role in normal oocyte development [54], as well as inducing increased intracellular calcium levels in sperm and can accelerate their hyperactivity [55]. Activated sperm are more likely to reach the oocyte, which is important for embryo formation [56]. Interestingly, *CAMK2G* was involved in calcium signaling pathways. *Calcium/Calmodulin dependent protein kinase II gamma* (*CAMK2G*) is known to be a protein that regulates apoptosis and acts as a downstream effector of *Na+, K+-ATPase* (*NKA*), serving as a regulator of cell adhesion, proliferation, and survival [57]. It was reported that *CAMK2G* plays a key role in mouse oocytes and controls their activation by restoring the cell cycle [58]. *ITGA7* is a member of the ECM–receptor interaction, and the protein encoded by this gene belongs to the integrin alpha chain family, which may direct cell migration, morphogenesis, differentiation, and metastasis [59,60]. As a novel target, *AKTIP* enriched in the ECM–receptor pathway, as an isoform of AKT (a serine/threonine kinase), has been widely researched in reproduction, including follicular development and embryo implantation through PI3K–Akt signaling [61,62]. Accumulating data indicate that ECM–receptor interactions also play a role in supporting and protecting cells during ovulation [63,64]. Our research preliminarily showed that the oviduct can alter the reproductive traits of goats through patent regulation of key pathways at the mRNA level. However, it is still unclear to a large extent whether there is a further connection between these key pathways and how they cooperate with each other, which needs further study.

Recently, several studies have confirmed that lncRNAs play key regulatory roles in goat reproduction [28,65]. Thus, the lncRNA–target gene (DEG) interaction network was constructed. The coexpression network showed that 71 DELs and 87 DEGs had synergistic effects, and most lncRNA–mRNA pairs were *trans*-regulated. In the high-fecundity group, the DELs (XLOC_021615, XLOC_119780, XLOC_076450) and their *trans*-regulated genes (*ATAD2*, *DEPDC5*, *TRPM6*) were significantly more abundant than those in the low-fecundity groups, and the trends of expression were consistent and were all related to embryo development. The target gene *ATAD2* of XLOC_021615 has been reported to be highly expressed in embryonic stem cells and was able to promote cell growth and chromatin templating activities (e.g., transcription) [66]. Knockout of DEPDC5, a trans-regulatory element of XLOC_119780, in mice caused malformed development of embryos [67], and *TRPM6*-deficient mice exhibited increased early embryonic mortality, which led to pregnancy failure [68]. Moreover, XLOC_020079, XLOC_107361, XLOC_169844, XLOC_252348, and the target genes (*ARHGEF2* and *RAPGEF6*) were mainly associated with prolificacy. In vitro, culture studies of mammalian oocytes reveal a well-developed increase in ARHGEF2 in mature oocytes, suggesting that ARHGEF2 may affect the development and maturation of oocytes [69]. In addition, *RAPGEF6* is essential for the high egg-laying rate of chickens [70]. Additionally, DE target genes, such as *cytoplasmic polyadenylation of element-binding protein 3 (CPEB3)**,* are RNA-binding proteins that bind to specific RNA sequences and affect the expression, cell localization, and stability of target RNAs after binding. CPEB3 knockout in mouse ovary can affect follicular development, and CPEB3 mutation in the ovary can inhibit granulosa cell proliferation and increase granulosa cell apoptosis [71]. It is worth noting that *CPEB3* was *trans*-regulated by XLOC_089239, XLOC_090063, XLOC_107409, XLOC_153574, XLOC_211271, and XLOC_251687, indicating that these lncRNAs might play key roles in cell proliferation or apoptosis by regulating the expression of *CPEB3*. We also found that the DE target gene PARD3, which was upregulated in high-fecundity goats, was simultaneously *trans*-regulated by XLOC_020079, XLOC_107361, XLOC_169844, and XLOC_252348 in the goat oviduct. PARD3 is usually involved in activating downstream TAZ transcription coactivators of the hippo pathway and functions as a cell growth promoter [72], indicating that lncRNA XLOC_020079, XLOC_107361, XLOC_169844, and XLOC_252348 could maintain proliferation in gamete cells. In conclusion, this study found that DELs may instead target genes and that DEGs coordinate the regulatory function of the oviduct, which may be the key factors regulating the kidding number of goats.

## 5. Conclusions

In summary, transcriptome oviduct research revealed differential regulation of mRNAs and lncRNAs associated with high and low fecundity in Yunshang black goats at the follicular phase. Based on the KEGG database, DEGs annotated a variety of physiological processes related to prolificacy traits, such as oocyte meiosis and the calcium, AMPK, and PI3K–Akt signaling pathways were enriched in the oviduct of goat with different kidding numbers. Furthermore, we predicted the target genes of DELs and compared them with DEGs to analyze the potential function of DELs. In addition, an interaction network of lncRNA genes was constructed, and several lncRNAs might be associated with prolificacy in high-fecundity goats were screened, which provided valuable resources for candidate lncRNAs in the oviduct. These DEG and DEL expression profiles provide a molecular mechanism for the increased prolificacy of goat breeding.

## Figures and Tables

**Figure 1 genes-13-01031-f001:**
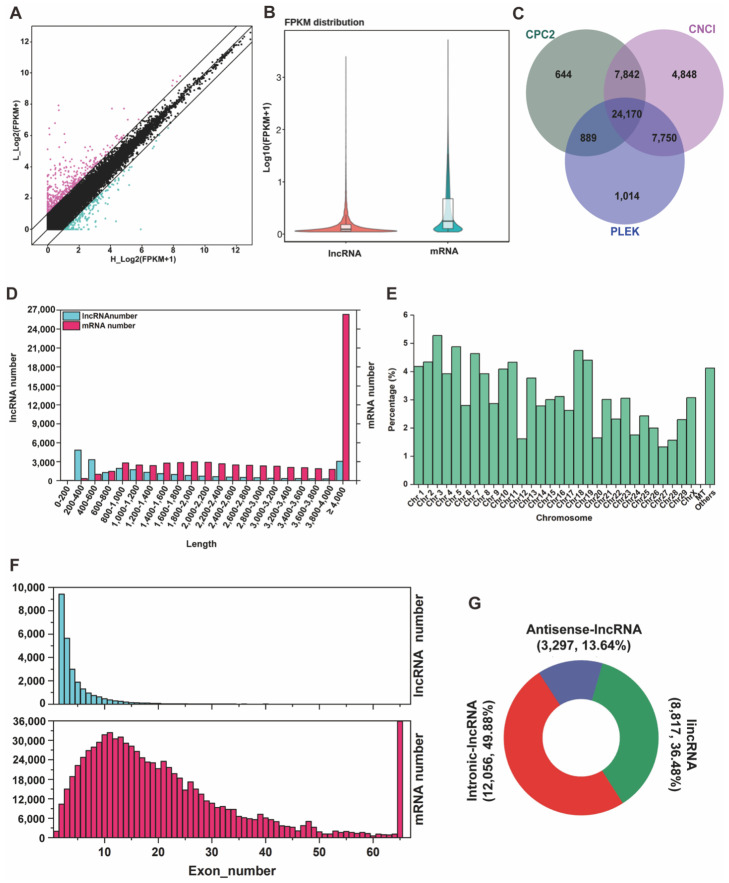
Expression profiles of lncRNA and mRNA in the oviduct. (**A**) The scatter plot presents the mRNA expression variations between FL and FH groups. The values of the *X* and *Y* axes represent the normalized mRNA signal values (log_2_ scaled), and the mRNAs above the top black line and below the bottom black line displayed greater than a two-fold change of up- and downregulation. **(B**) The expression level of lncRNA transcripts and mRNA transcripts. (**C**) Venn diagram showing the common and unique number of novel lncRNAs by CNCI, CPC2, and PLEK methods. (**D**) The length statistics of lncRNA and mRNA. (**E**) The distribution of lncRNAs in different chromosomes. (**F**) The statistics of lncRNA and mRNA exon number. (**G**) Classification of novel lncRNAs, including lincRNAs, intronic-lncRNAs, and antisense-lncRNAs.

**Figure 2 genes-13-01031-f002:**
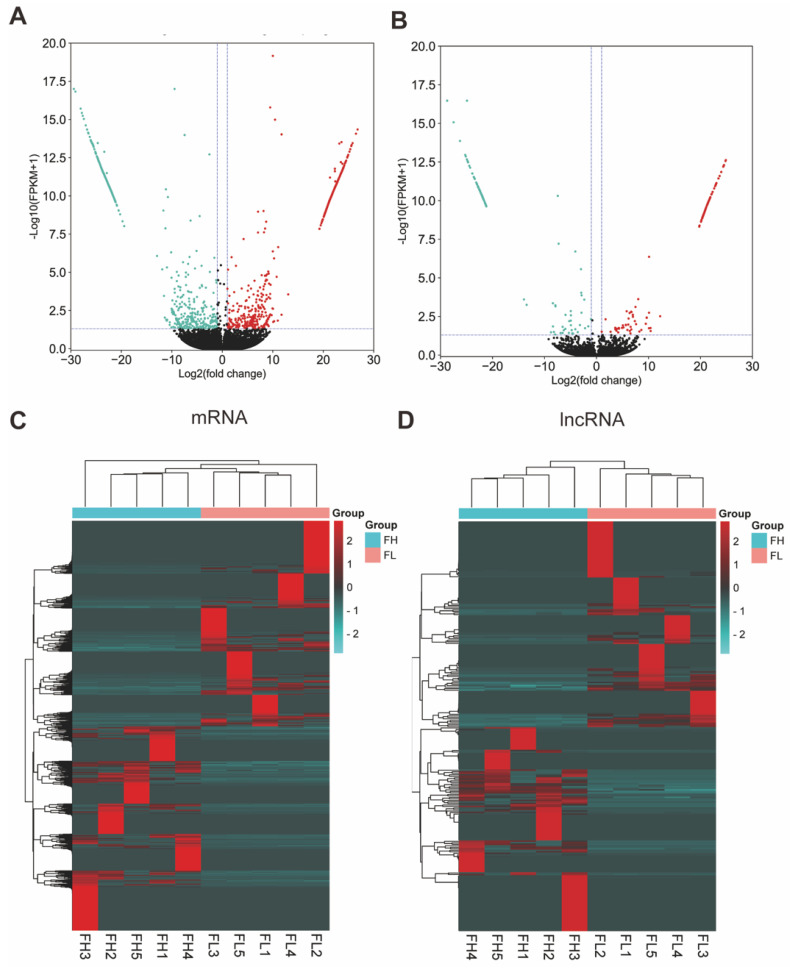
The analysis of differentially expressed mRNAs (DEGs) and differentially expressed lncRNAs (DELs). (**A**) Volcano plot of DEGs and (**B**) DELs in FL vs. FH. The vertical lines correspond to | log2 FC (fold change) | > 1 and in up–regulation or down–regulation; the horizontal line represents *q*-value < 0.05; green points refer to down–regulated and red points refer to up–regulated. (**C**) Hierarchical cluster analysis of DEGs and (**D**) DELs in FL vs. FH. The color scale indicates log2(FPKM) and intensity increases from green to red, which indicates down– and up–regulation, respectively.

**Figure 3 genes-13-01031-f003:**
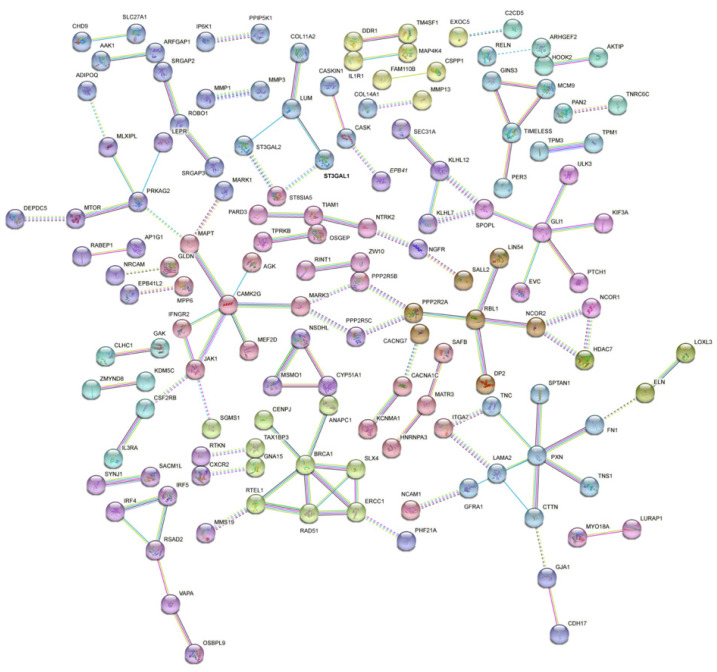
Protein–protein interaction (PPI) network was constructed by the STRING database using protein-coding genes in FL vs. FH. The minimum required interaction score was set as 0.9.

**Figure 4 genes-13-01031-f004:**
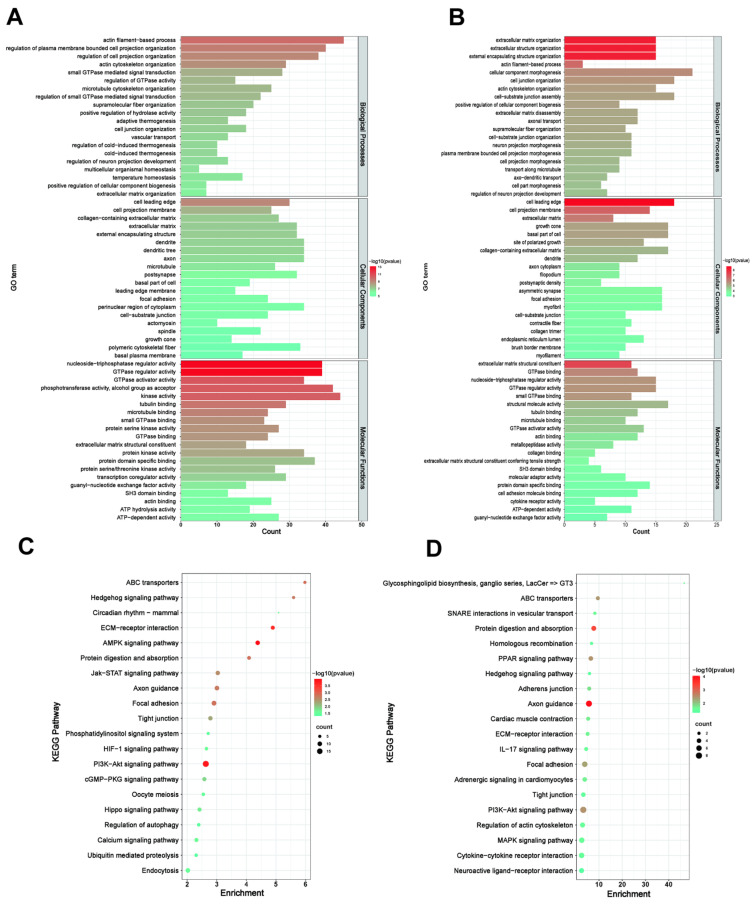
Gene Ontology (Go) annotation and Kyoto Encyclopedia of Genes and Genomes (KEGG) pathway enrichment of differentially expressed mRNAs, (**A**) GO and (**C**) KEGG analysis of DEGs in FL vs. FH. (**B**) GO and (**D**) KEGG analysis of target genes of DELs.

**Figure 5 genes-13-01031-f005:**
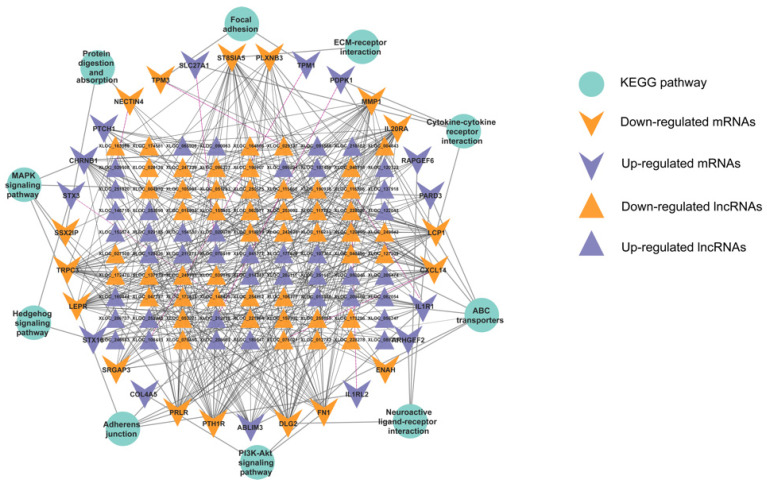
The interaction networks of “pathways-differentially expressed lncRNAs (DELs)-target genes”. The solid and dotted lines are present for trans- and cis-regulation functions, respectively. The triangle is for lncRNA, the “V” represents the target gene, and the ellipse is for the KEGG pathway of the target genes, respectively. A total of 90 DELs were cis or trans-regulated with 31 mRNAs in the interactive network.

**Figure 6 genes-13-01031-f006:**
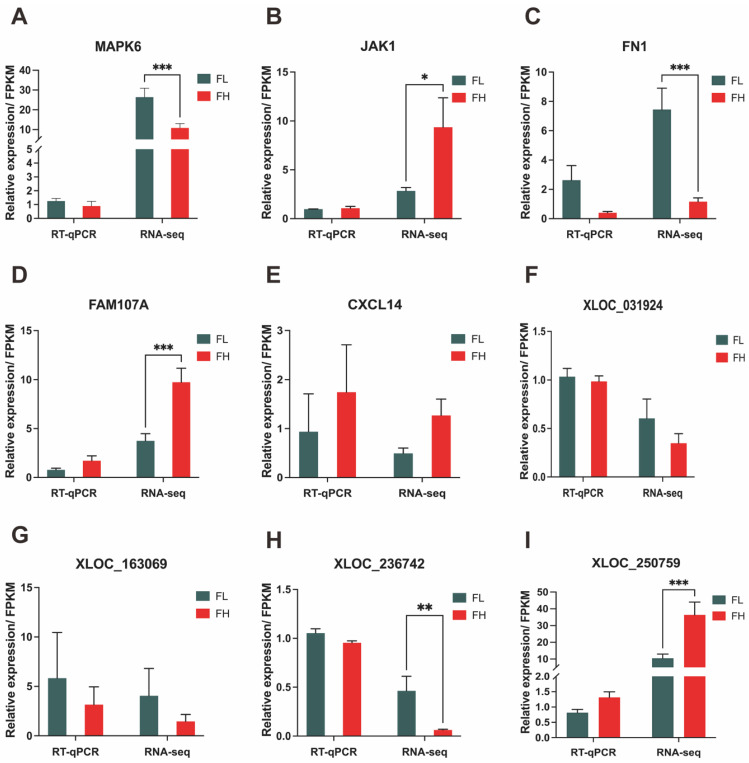
Validation of the RNA-seq data by RT-qPCR. The RT-qPCR data are presented as relative gene expression. RNA-seq data are presented as fragments per kilobase of transcripts per million mapped reads (FPKM). (**A**–**E**) Selected mRNAs and (**F**–**I**) lncRNAs were validated by RT-qPCR and RNA-seq, respectively. *** *p* < 0.001, ** *p* < 0.01, * *p* < 0.05.

**Table 1 genes-13-01031-t001:** The information of RNA-seq data.

Sample	Clean Reads	Clean Base (bp)	Length	Q20 (%)	Q30 (%)	GC (%)	Total Mapped (%)
FL-1	52,752,779	15,825,833,700	150	98.35	94.75	51.45	97.13
FL-2	60,824,479	18,247,343,700	150	98.15	94.60	50.70	96.86
FL-3	53,704,715	16,111,414,500	150	98.20	94.60	50.10	96.86
FL-4	51,703,403	15,511,020,900	150	98.45	95.05	50.10	97.35
FL-5	54,659,450	16,397,835,000	150	98.35	95.00	48.55	96.40
FH-1	56,457,768	16,937,330,400	150	98.50	95.25	51.15	96.90
FH-2	51,119,834	15,335,950,200	150	98.08	94.20	50.00	96.40
FH-3	67,474,481	20,242,344,300	150	97.80	93.50	51.15	96.99
FH-4	52,753,848	15,826,154,400	150	98.25	94.40	49.25	97.18
FH-5	50,538,996	15,161,698,800	150	98.50	95.10	51.25	97.18

## Data Availability

Not applicable.

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
