# Peer review of "Integrated Analysis of mRNAs and Long Non-Coding RNAs Expression of Oviduct That Provides Novel Insights into the Prolificacy Mechanism of Goat (Capra hircus)"

_genes, 2022, doi:10.3390/genes13061031_

Round 1

Reviewer 1 Report

The manuscript “Integrated Analysis of mRNAs and Long Non-Coding RNAs 2 Expression of Oviduct That Provides Novel Insights into the 3 Prolificacy Mechanism of Goat (Capra hircus)” analyses oviducts from does with low or high fertility (average litter size 1.8±0.27 and 3.4±0.42, respectively) extracting RNA to find genes related to prolificacy. The aim of the research is interesting and the materials and methods adequate, well explained, and relevant. Nonetheless, the manuscript writing is deficient, there are many grammar and translation mistakes that make it very difficult to grasp the meaning of several sentences. Also, some terms are not adequate, the term “she-goat” should be replaced with “doe”. Furthermore, the discussion needs to be restructured, it is poorly written, and the ideas presented are not linked to each other and the findings. Likewise, the conclusion is not in line with the aim and title of the research, as it doesn’t present what is concluded regarding the differences between low or high prolificacy in goats.

Author Response

Point 1:

The manuscript writing is deficient, there are many grammar and translation mistakes that make it very difficult to grasp the meaning of several sentences.

Response 1:

Thanks for raising this important issue, we have contacted a professional company to correct our manuscript.

Point 2:

Some terms are not adequate, the term “she-goat” should be replaced with “doe”

Response 2:

We are very sorry for our incorrect writing, so we revised it in the article:

It can not only screen out candidate coding genes related to doe fertility but also identify key reproductive regulatory factors of the oviduct.

Adult doe, with the same weights and hearths, were raised with free access to feed and water under the same temperature.

Point 3:

The discussion needs to be restructured, it is poorly written, and the ideas presented are not linked to each other and the findings. Likewise, the conclusion is not in line with the aim and title of the research, as it doesn’t present what is concluded regarding the differences between low or high prolificacy in goats.

Response 3:

We thank the raising this question, so we have adjusted the discussion and conclusions in detail as follows:

Discussion

Clarifying the regulatory mechanism of the kidding number can provide a dynamic theoretical basis for goat breeding. Although the ovary determines the number of ovulations, the oviduct, as a conduit between the ovary and uterus, plays important roles in gamete storage, maturation, fertilization, and early embryonic development [41-43]. It largely determines the success of reproduction in mammals. Previous studies on oviduct function have been carried out in humans [41], bovines [44], and porcines [17], but they are limited to the study of mRNA. Nonetheless, the roles of lncRNAs in oviduct reproduction remain largely unknown. Therefore, the data mining of critical genes and the regulatory mechanisms of non-coding RNAs in goat prolificacy traits is still a large research topic. Herein, RNA-Seq was used to analyze the transcriptome expression profiles of oviduct mRNA and lncRNA in Yunshang black goats with low- and high-fertility during the follicular phase.

In this study, the functions of lncRNA and the target genes of lncRNAs were revealed from the perspective of the oviduct in the follicular phase. A total of 68,149 mRNAs and 25,958 lncRNAs were identified in the goat oviduct, of which 24,170 novel lncRNAs and 1,788 known lncRNA transcripts were obtained. The results of sequence analyses showed that the GC content was 48.55~51.45%, whereas the Q30 content was 93.50~95.25% by base quality and composition analysis, indicating that high-quality libraries and sequencing were applied. Many studies have shown that lncRNAs have unique characteristics compared with mRNAs; for example, the transcripts of lncRNAs tend to be shorter in length, have fewer exons, and have lower in the expression levels [45, 46]. These characteristics were consistent with this research. In addition, compared with the 4,926 and 11,723 lncRNAs found in the ovary [26] and uterus [27] of goat, respectively, there are more lncRNAs in the oviduct, but their exons are similar, which reflects the specific expression patterns and characteristics of the oviduct. At the same time, most lncRNAs were widely located on chromosomes 3 (NC_030810.1), 5 (NC_030812.1), and 18 (NC_030823.1), which is inconsistent with the distribution in other reproductive organs of goats. All the results explained that lncRNAs in particular were reliable in the oviducts of goats.

Herein, we identified a wealth of GO terms and KEGG pathways. Subsequently, we analyzed the important pathways and pertinent DEGs, which are involved in reproductive regulation, including AMPK, PI3K-Akt, cGMP-PKG, HIF-1, Jak-STAT, and calcium signaling pathway, ECM-receptor interaction, ABC transporters, protein digestion and absorption, focal adhesion, tight junction, circadian rhythm, oocyte meiosis, and ubiquitin-mediated proteolysis. Reproduction is a function that requires energy and can only be carried out when mammals have enough available energy. The AMPK signaling pathway serves as a cellular energy receptor that responds to low levels of ATP, and AMPK activation affects gene expression in biological processes such as embryonic development, spermatocyte DNA damage repair and gonadal steroid hormone production, suggesting that AMPK activation affects the transcriptional regulation of reproductive processes [47]. The follicular phase is a key period of follicular development, and the normal development of follicles and ovulation is an important reason for high fecundity. In the reproductive system, the PI3K-Akt signaling pathway plays important roles in regulating cell metabolism, as well as in the survival and activation of primordial follicles [48, 49]. The oviduct plays active roles in the processes of oocyte maturation, migration, and fertilization, which are important in reproduction and can directly determine the kidding number. Some studies have pointed out that the interaction between oocytes and oviducts in mammals has a positive impact on oocyte maturation and can restore the oocyte meiosis process [50]. Notably, the focal adhesion and tight junction pathways were significantly enriched in the oviduct in the follicular phase, which are important functions that determine fertilization [51]. Our study showed that these DEGs, including ABCC9 and APC, were upregulated in the FL vs. FH groups and were involved in ABC transporters and regulation of the actin cytoskeleton, respectively. It has been reported that ABCC9, a molecule that matches the K (+)-selective pores, regulates the ABC transporter protein of K (+) channels by assembling ATP-sensitive K (+) channels in smooth muscle and transverse muscle [52]. Fertilization occurs primarily in the ampulla of the oviduct, which has an extensively folded epithelial layer and is collated with a thin layer of smooth muscle [43]. Hence, we hypothesize that ABCC9contributes to the contraction of oviductal epithelial tissue muscles, thereby aiding gamete transport and contributing to complete fertilization. Meanwhile, APC can promote cell proliferation, differentiation, migration and apoptotic [53]. In the oviduct, the calcium signaling pathway plays an important role in normal oocyte development [54], as well as inducing increased intracellular calcium levels in sperm and can accelerate their hyperactivity [55]. Activated sperm are more likely to reach the oocyte, which is important for embryo formation [56]. Interestingly, CAMK2G was involved in calcium signaling pathways. Calcium/Calmodulin dependent protein kinase II gamma (CAMK2G) is known to be a protein that regulates apoptosis and acts as a downstream effector of Na+, K+-ATPase (NKA), serving as a regulator of cell adhesion, proliferation, and survival [57]. It was reported that CAMK2G plays a key role in mouse oocytes and controls their activation by restoring the cell cycle [58]. ITGA7 is a member of the ECM-receptor interaction, and the protein encoded by this gene belongs to the integrin alpha chain family, which may direct cell migration, morphogenesis, differentiation, and metastasis [59,60]. As a novel target, AKTIP enriched in the ECM-receptor pathway, as an isoform of AKT (a serine/threonine kinase), has been widely researched in reproduction, including follicular development and embryo implantation through PI3K-AKT signaling [61,62]. Accumulating data indicate that ECM-receptor interactions also plays a role in supporting and protecting cells during ovulation [63,64]. Our research preliminarily showed that the oviduct can alter the reproductive traits of goats through patent regulation of key pathways at the mRNA level. However, it is still unclear to a large extent whether there is a further connection between these key pathways and how they cooperate with each other, which needs further study.

Recently, several studies have confirmed that lncRNAs play key regulatory roles in goat reproduction [28,65]. Thus, the lncRNA-target gene (DEG) interaction network was constructed. The coexpression network showed that 71 DELs and 87 DEGs had synergistic effects, and most lncRNA-mRNA pairs were trans-regulated.In the high-fecundity group, the DELs (XLOC_021615, XLOC_119780, XLOC_076450) and their trans-regulated genes (ATAD2, DEPDC5, TRPM6) were significantly more abundant than those in the low-fecundity groups, and the trends of expression were consistent and were all related to embryo development. The target gene ATAD2 of XLOC_021615 has been reported to be highly expressed in embryonic stem cells and was able to promote cell growth and chromatin templating activities (e.g., transcription) [66]. Knockout of DEPDC5, a trans-regulatory element of XLOC_119780, in mice caused malformed development of embryos [67], and TRPM6-deficient mice exhibited increased early embryonic mortality, which led to pregnancy failure [68]. Moreover, XLOC_020079, XLOC_107361, XLOC_169844, XLOC_252348 and the target genes (ARHGEF2 and RAPGEF6) were mainly associated with prolificacy. In vitro, culture studies of mammalian oocytes reveal a well-developed increase in ARHGEF2 in mature oocytes, suggesting that ARHGEF2 may affect the development and maturation of oocytes [69]. In addition, RAPGEF6 is essential for the high egg-laying rate of chickens [70]. Additionally, DE target genes, such as cytoplasmic polyadenylation of element-binding protein 3 (CPEB3), are RNA-binding proteins that bind to specific RNA sequences and affect the expression, cell localization and stability of target RNAs after binding. CPEB3 knockout in mouse ovary can affect follicular development, and CPEB3 mutation in ovary can inhibit granulosa cell proliferation and increase granulosa cell apoptosis [71]. Note to worth, CPEB3 was trans regulated by XLOC_089239, XLOC_090063, XLOC_107409, XLOC_153574, XLOC_211271, and XLOC_251687, indicating that these lncRNAs might play key roles in cell proliferation or apoptosis by regulating the expression of CPEB3. We also found that the DE target gene PARD3, which was upregulated in high-fecundity goats, was simultaneously trans-regulated by XLOC_020079, XLOC_107361, XLOC_169844, and XLOC_252348 in the goat oviduct. PARD3 is usually involved in activating downstream TAZ transcription coactivators of the hippo pathway and functions as a cell growth promoter [72], indicating that lncRNA XLOC_020079, XLOC_107361, XLOC_169844, and XLOC_252348 could maintain proliferation in gamete cells. In conclusion, this study found that DELs may instead target genes and that DEGs coordinate the regulatory function of the oviduct, which may be the key factors regulating the kidding number of goats.

Conclusion

In summary, transcriptome oviduct research revealed differential regulation of mRNAs and lncRNAs associated with high and low fecundity in Yunshang black goats at the follicular phase. Based on the KEGG database, DEGs annotated a variety of physiological processes related to prolificacy trait, such as oocyte meiosis and the calcium, AMPK and PI3K-Akt signaling pathways were enriched in the oviduct of goat with different kidding numbers. Furthermore, we predicted the target genes of DELs and compared them with DEGs to analyze the potential function of DELs. In addition, an interaction network of lncRNA-genes was constructed, and several lncRNAs might be associated with prolificacy in high fecundity goats were screened, which provided valuable resources for candidate lncRNAs in the oviduct. These DEG and DEL expression profiles provide a molecular mechanism for the increased prolificacy of goat breeding.

Reviewer 2 Report

The manuscript is well written and has good scientific quality. It can only be improved in the introduction and conclusion sections. Please refer to the comments given in the text of the attached manuscript.

Author Response

Point 1:

Amendments to the Introduction section.

Response 1:

Thanks to the reviewer for helpful comments on the importance of the goat industry worldwide, which has led to a further understanding of goats and highlighted the importance of this research. According to your suggestion, we added references 1, 2, 3, 4, 5, 6, 31, 32, which are as follows:

Goat farming is practiced worldwide, with goat products having a favorable image [1]. Goat production is one of the key elements contributing to the economy of farmers living in arid and semiarid regions [2,3]. The number of goats has increased globally, even in countries with high and intermediate incomes [4]. Worldwide, 96% of the milk- and meat-producing goat populations are found in developing countries, and 4% are found in developed countries [5,6].

Increasing meat production using scientific, accurate, and precise selective programs are one of the most important goals for genetic improvement of goats [31,32].

  1. Gooki, F.G.; Mohammadabadi, M.R.; Fozi, M.A.; Soflaei, M. Association of biometric traits with growth hormone gene diversity in Raini cashmere goats. Walailak Journal of Science and Technology. 2019, 16, 499-508.
  2. Askari, N.; Mohammad, M.R.; Baghizadeh, A. ISSR markers for assessing DNA polymorphism and genetic characterization of cattle, goat and sheep populations. Iranian Journal of Biotechnology. 2011, 9, 222-229.
  3. Barazandeh, A.; Mohammadabadi, M.R.; Ghaderi-Zefrehei, M.; Rafeie, F.; Imumorin, I.G. Whole genome comparative analysis of CpG islands in camelid and other mammalian genomes. Mammalian Biology. 2019, 98, 73-79.
  4. Gholamhoseini F.G.; Mohammadabadi M.R.; Asadi Fozi M. Polymorphism of the growth hormone gene and its effect on production and reproduction traits in goat. Iranian Journal of Applied Animal Science. 2018, 8, 653-659.
  5. Noori, A.N.; Behzadi, M.; Mohammadabadi, M.R. Expression pattern of Rheb gene in Jabal Barez Red goat. The Indian journal of animal sciences. 2017, 87, 1375-1378.
  6. Mohammadabadi, M.; Bordbar, F.; Jensen, J.; Du, M.; Guo, W. Key genes regulating skeletal muscle development and growth in farm animals. Animals. 2021, 11, 835.
  7. Molaei Moghbeli, S.; Barazandeh, A.; Vatankhah, M.; Mohammadabadi, M. Genetics and non-genetics parameters of body weight for post-weaning traits in Raini cashmere goats. Trop Anim Health Prod. 2013, 45, 1519-1524.
  8. Mohammadabadi M.R.; Asadollahpour H. Leptin gene expression in Raini Cashmere goat using Real Time PCR. Agricultural Biotechnology Journal. 2021, 13, 197-214.

Point 2:

Amendments to the Conclusion section.

Response 2:

We have restructured the Conclusions section and the results are as follows:

In summary, transcriptome oviduct research revealed differential regulation of mRNAs and lncRNAs associated with high and low fecundity in Yunshang black goats at the follicular phase. Based on the KEGG database, DEGs annotated a variety of physiological processes related to prolificacytrait, such as oocyte meiosis and the calcium, AMPK and PI3K-Akt signaling pathways were enriched in the oviduct of goat with different kidding numbers. Furthermore, we predicted the target genes of DELs and compared them with DEGs to analyze the potential function of DELs. In addition, an interaction network of lncRNA-genes was constructed, and several lncRNAs might be associated with prolificacy in high fecundity goats were screened, which provided valuable resources for candidate lncRNAs in the oviduct. These DEG and DEL expression profiles provide a molecular mechanism for the increased prolificacy of goat breeding.

Round 2

Reviewer 1 Report

All previous observations have been met and thus, the article has been greatly improved and I consider it to be suitable for publication. The changes made to the discussion and conclusion are adequate and more in line with the research. It is advisable to do a last spelling and grammar check for any mistakes that might have been missed.